# Interleukin-17 is disease promoting in early stages and protective in late stages of experimental periodontitis

Anneke Wilharm[1]☯, Christoph Binz[1]☯*, Inga Sandrock[1], Francesca Rampoldi[1],
Stefan Lienenklaus[2], Eva Blank[3,4], Andreas Winkel[3,4], Abdi Demera[1], Avi-Hai Hovav[5],
Meike Stiesch[3,4], Immo Prinz[1,6]

1 Institute of Immunology, Hannover Medical School, Hannover, Germany, 2 Institute of Laboratory Animal
Science, Hannover Medical School, Hannover, Germany, 3 Department of Prosthetic Dentistry and
Biomedical Materials Science, Hannover Medical School, Hannover, Germany, 4 Lower Saxony Centre for
Biomedical Engineering, Implant Research and Development (NIFE), Hannover, Germany, 5 Institute of
Dental Sciences, Faculty of Dental Medicine, Hebrew University, Jerusalem, Israel, 6 Institute of Systems
Immunology, University Medical Center Hamburg-Eppendorf, Hamburg, Germany

☯ These authors contributed equally to this work.
* christoph.binz@arcor.de

doi.org/10.1371/journal.pone.0265486

Medical Sciences, INDIA

**Data Availability Statement:** All relevant data are
within the manuscript and its Supporting
Information files.

## Abstract

Periodontitis is one of the most common infectious diseases in humans. It is characterized
by a chronic inflammation of the tooth-supporting tissue that results in bone loss. However,
the role and source of the pro-inflammatory cytokine interleukin-17 (IL-17) and of the cells
producing it locally in the gingiva is still controversial. Th17 αβ T cells, CD4+ exFoxP3+ αβ T
cells, or IL-17-producing γδ T cells (γδ17 cells) seem to be decisive cellular players in peri-
odontal inflammation. To address these issues in an experimental model for periodontitis,
we employed genetic mouse models deficient for either γδ T cells or IL-17 cytokines and
assessed the bone loss during experimental periodontal inflammation by stereomicroscopic,
histological, and μCT-analysis. Furthermore, we performed flow-cytometric analyses and
qPCR-analyses of the gingival tissue. We found no γδ T cell- or IL-17-dependent change in
bone loss after four weeks of periodontitis. Apart from that, our data are complementary with
earlier studies, which suggested IL-17-dependent aggravation of bone loss in early peri-
odontitis, but a rather bone-protective role for IL-17 in late stages of experimental periodonti-
tis with respect to the osteoclastogenicity defined by the RANKL/OPG ratio.

## Introduction

γδ T cells are innate T cells developing in embryonic and postnatal phase [1], conserved
among almost all jawed vertebrates [2] including mice and humans. In mice, γδ T cells are
known to constitute the majority of epithelial-resident T cells [3]. After their development in
the thymus, they egress in several waves of cells expressing distinct γδTCRs to home to these
epithelial tissues [1, 4], as for example the skin and the gingiva [3, 5]. The dermal layer of the
skin is populated by Vγ4+ and Vγ6+ γδ T cells that rapidly produce IL-17 to induce

**Funding:** This work was supported by the German–Israeli Foundation for Scientific Research and Development Grant 1432 (to A.-H.H. and I.P., http://www.gif.org.il/pages/default.aspx), by a grant from the Niedersächsisch-Israelische Forschungsförderung (to A.-H.H. and I.P.), and by the DG PARO CP GABA (to A.W., I.P., and M.S.) A. Wilharm and C. Binz were supported by the Hannover Biomedical Research School. The funders had no role in study design, data collection and analysis, decision to publish, or preparation of the manuscript.

**Competing interests:** The authors have declared that no competing interests exist.

inflammation [3, 6, 7]. Recently we and others found that also the gingiva, especially the junctional epithelium (JE) directly lining the teeth, is populated by mainly Vγ6+ γδ T cells [8].

With a prevalence of almost 50% in adults, periodontitis is a very common [9, 10] inflammatory disease of the tooth-supporting tissue. It is induced by oral bacteria as well as by the subsequent reaction of the host immune system [11]. Untreated it may lead to tooth-loss, since the chronic inflammation is accompanied by the loss of periodontal attachment and retraction of the alveolar bone [12]. *Porphyromonas gingivalis*, together with *Tennerella forsythia* and *Treponema denticola*, belongs to the "red complex" of bacteria, which are strongly associated with periodontal disease, originally described by Socransky *et al.* [13]. The invading oral bacteria can be disseminated to other sites of the organism [14] and drive inflammation there [15, 16], which was already proposed in 1891 in the aptly named article "The human mouth as a focus of infection" [17]. Furthermore, periodontitis is linked to diseases like cardiovascular diseases [18] and rheumatoid arthritis [19, 20].

During the course of periodontitis, osteoclasts in the tooth-supporting tissue are activated by "receptor activator of NFκB ligand" (RANKL) [21] and "secreted osteoclastogenic factor of activated T cells" (SOFAT) [22] resulting in the destruction of alveolar bone. RANKL expression has been reported to be induced on osteoblastic and ligament cells by IL-17 produced by T cells [23, 24] as well as being expressed by those T cells themselves, both also true for periodontitis [14]. SOFAT, a RANKL-independent activator of osteoclastogenesis, is directly secreted from T cells and increased in periodontitis patients.

Overall, IL-17 (IL-17A) is known as a pro-inflammatory cytokine. The corresponding IL-17 receptor IL-17RA is expressed by macrophages or dendritic cells (DC), as well as by fibroblasts or epithelial cells and its engagement leads to secretion of cytokines, matrix-metalloproteinases or antimicrobial peptides [25, 26]. Thus, also in the gingiva, IL-17 attracts especially neutrophils, which is impaired in IL-17RA knockout mice aggravating the alveolar bone loss during periodontitis [27].

Nonetheless, the context of action seems to be important when judging its effects as beneficial or detrimental [28–30]. For example, IL-17 production by γδ T cells (γδT17) was shown to be important for effective bacterial clearance [31–41] or control of fungal burden [42, 43]. In the oral cavity, Vγ6+ γδT17 play a protective role in immunity against *Candida albicans* in the tongue epithelium [5]. For induced orthodontic tooth-movement, they were described as beneficial, since their presence allows bone-remodeling in the tooth-supporting tissue [44]. However, γδT17 cells are associated with tissue destruction in autoimmune diseases in a variety of tissues. In psoriatic skin, γδT17 cells have been described as important pro-inflammatory populations [28, 45] and depletion led to amelioration of the symptoms [46]. In models of spondylo- and rheumatoid arthritis, γδT17 cells were reported to be a major source of IL-17 and thus to aggravate experimental enthesitis [47–49]. Although Th17 cells are also important producers of IL-17 in experimental autoimmune encephalitis (EAE), the critical role of γδT17 cells in induction of this pathology is emerging [50–53].

Still, it is unclear whether exFoxp3 Th17 αβ T cells [14] or amphiregulin producing γδ T cells [54], that were shown to produce IL-17 in the gingiva [8], play the most decisive role for the outcome of the periodontal inflammation. Moreover, after ten days of inflammation, bone loss was reduced in *IL17af*−/− mice [14], but enhanced in IL-17RA knockout mice after six weeks of inflammation [27]. Furthermore, at other barrier sites, γδ T cells have been reported to be beneficial with respect to body-barrier integrity [55, 56], host-defense [5] or host-commensal homeostasis [57]. Therefore, it is unclear whether IL-17 and its producers have a bone-protective or a bone-damaging role. It has been proposed that IL-17 is a "double-edged sword" [14] in this context, fighting pathogens but simultaneously damaging the bone-supporting tissue.

To shed more light onto these partially contradicting results, we used an experimental model of ligature-induced periodontitis and analyzed the ligature-induced bone loss in *Tcrd-*

H2BeGFP (as WT-control expressing eGFP in all γδ T cells), $Tcrd^{-/-}$ and $IL17af^{-/-}$ mice after four weeks of inflammation was analyzed, representing a time point between the two assessed ones in the publications mentioned above [14, 27, 54]. Surprisingly, we could not detect any differences between those three genotypes. Therefore, we conclude that the IL-17 production by the present αβ T cells must be more relevant during the course of periodontitis, than the one by γδ T cells, as previously shown by Tsukasaki *et al*. and that IL-17 indeed acts as a double-edged sword. Moreover, our data add to the idea that the bone-damaging effects of IL-17 in periodontitis might be an evolutionarily beneficial emergency-stop switch for the inflammation, that otherwise persists for a longer period and threatens the whole organism.

## Results

### *P. gingivalis* is dispensable for bone loss in ligature-induced periodontitis in mice

Since there are different methods in use to assess the bone erosion in periodontitis experimentally [58–60], we aimed to investigate the bone erosion in *Tcrd*-H2BeGFP mice, $Tcrd^{-/-}$ mice and $IL17af^{-/-}$ mice performing two slightly distinct models and applying two different ways of analysis afterwards. The first approach was based on an initial reduction of oral microbiota by application of ampicillin and kanamycin for five days. After two days without antibiotics, ligatures were placed, tying a piece of suture material around both second molars of the maxilla and fixing it with two nods on the lingual side of the teeth. *P. gingivalis* was subsequently administered orally five times per week for four consecutive weeks (Fig 1A). On the other hand, we only applied a ligature around the second molars of the mice, sacrificed and analyzed the mice 28 days later. The analysis of bone erosion was performed after imaging the jaws by measuring the cemento-enamel-junction to alveolar-bone-crest (CEJ-ABC) distance at defined tooth-structures [58] at the ligated second molar or by measuring the area of the teeth laying open between CEJ and ABC (Fig 1B). Both models induced significant and comparable bone-loss (Fig 1C and 1D). There seemed to be a more severe bone erosion in the *Tcrd*-H2BeGFP mice compared to the $Tcrd^{-/-}$ and $IL17af^{-/-}$ mice, which showed very similar amounts of bone loss, but no statistically significant differences were detectable between the three genotypes. Furthermore, we analyzed the immune cells in the gingiva of the respective mice by flow cytometry (full gating strategy in S1 Fig). The number of CD45+ cells or of T cells did not increase in response to either treatment (Fig 2A). Looking into the T cell compartment we could not detect any significant changes induced by the treatment regarding αβ T cells, γδ T cells or the predominant Vγ6+ γδ T cells in percentagewise or absolute presence in the gingiva, despite there seemed to be a slight tendency towards an increase in both subsets for the ligature and ligature plus *P. gingivalis* treatment group (Fig 2B and 2C), confirming earlier reports [14]. While neutrophils in ligature treated animals, in line with the literature [14, 27], seemed to increase percentagewise and showed a statistically significant increase in absolute numbers, the macrophages showed a significant percentagewise decrease and no absolute decrease, suggesting an enhanced neutrophil presence in the gingiva (Fig 2D). Due to the high similarity of the outcomes of both models, we decided to restrict the following experiments to the ligature-model.

### RT-PCR reveals protective role of IL-17 in periodontitis independent of γδ T cells

The bone loss observed in the model used here, is a result of the inflammation-associated T cell activation in the gingiva, leading to RANKL secretion by T cells, that in turn binds to its receptor RANK on osteoclasts inducing their final differentiation and thereby intensifying the

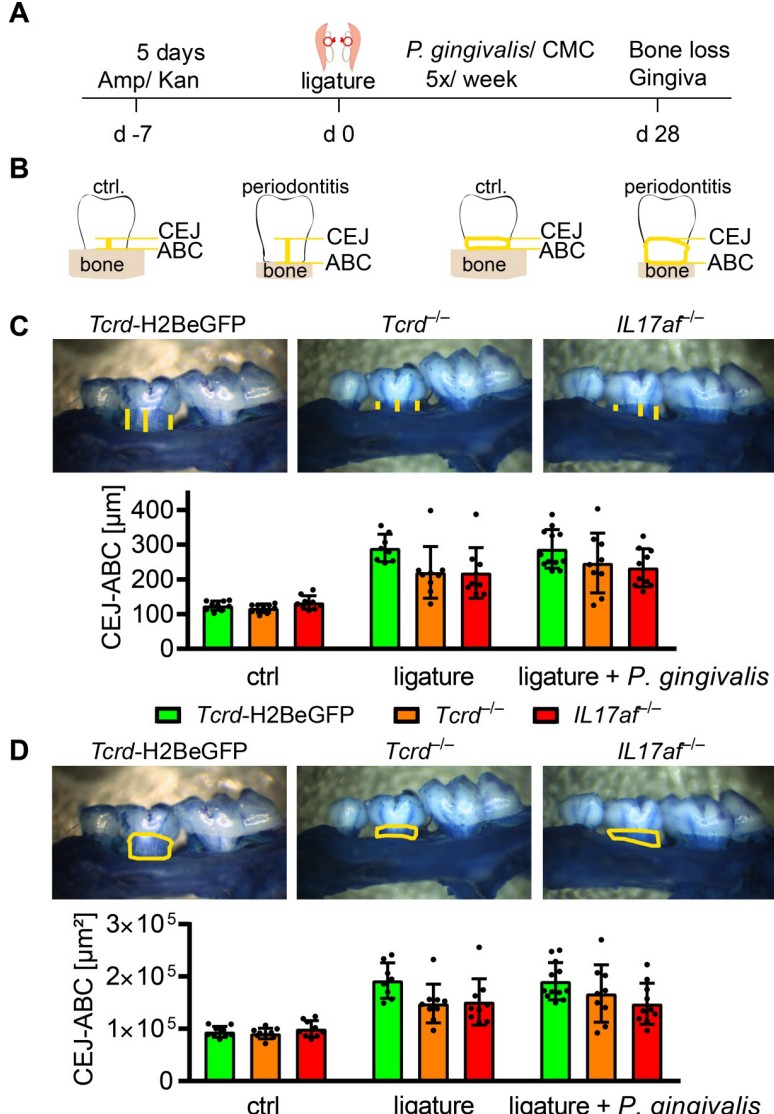

**Fig 1. No differences in ligature-induced bone-loss between *Tcrd*-H2BeGFP, *Tcrd*<sup>−/−</sup> and *IL17af*<sup>−/−</sup> mice with or without presence of *P. gingivalis*.** (A) scheme of the procedure applied to induce periodontitis. (B) Scheme of the bone-loss-analysis by measuring the CEJ-ABC distance (left) or the area laying open between CEJ and ABC (right). (C) CEJ-ABC distance and representative raw-data of defleshed jaws after methylene blue staining. Each point represents a mouse. Data were statistically analyzed by Kruskal-Wallis test, mean and SD are shown. (D) CEJ-ABC area and representative raw-data of defleshed jaws after methylene blue staining. Each point represents a mouse. Data were obtained from three independent experiments and statistically analyzed by Kruskal-Wallis test and Dunn´s multiple comparisons, mean and SD are shown. A p-value below 0.05 was considered as significant. All genotypes show statistically significant bone loss compared to control, which is not depicted in the figure.

bone loss [21]. A RANKL-inhibitory protein is osteoprotegerin (OPG), that is secreted by osteo-blasts and blocks the RANKL-RANK binding by binding to RANKL [21]. Besides a qPCR-measurement of their induction during the experimental periodontitis, we assessed the osteo-clast-activity based on the induction of the tartrate-resistant acid-phosphatase (TRAP), an enzyme responsible for bone resorption by osteoclasts [61] and growth-arrest-specific-6 (GAS6), that has been reported to be a key factor for establishment of oral homeostasis [62] and that activates osteoclasts via the Tyrosine-protein-kinase-receptor Tyro-3 [63].

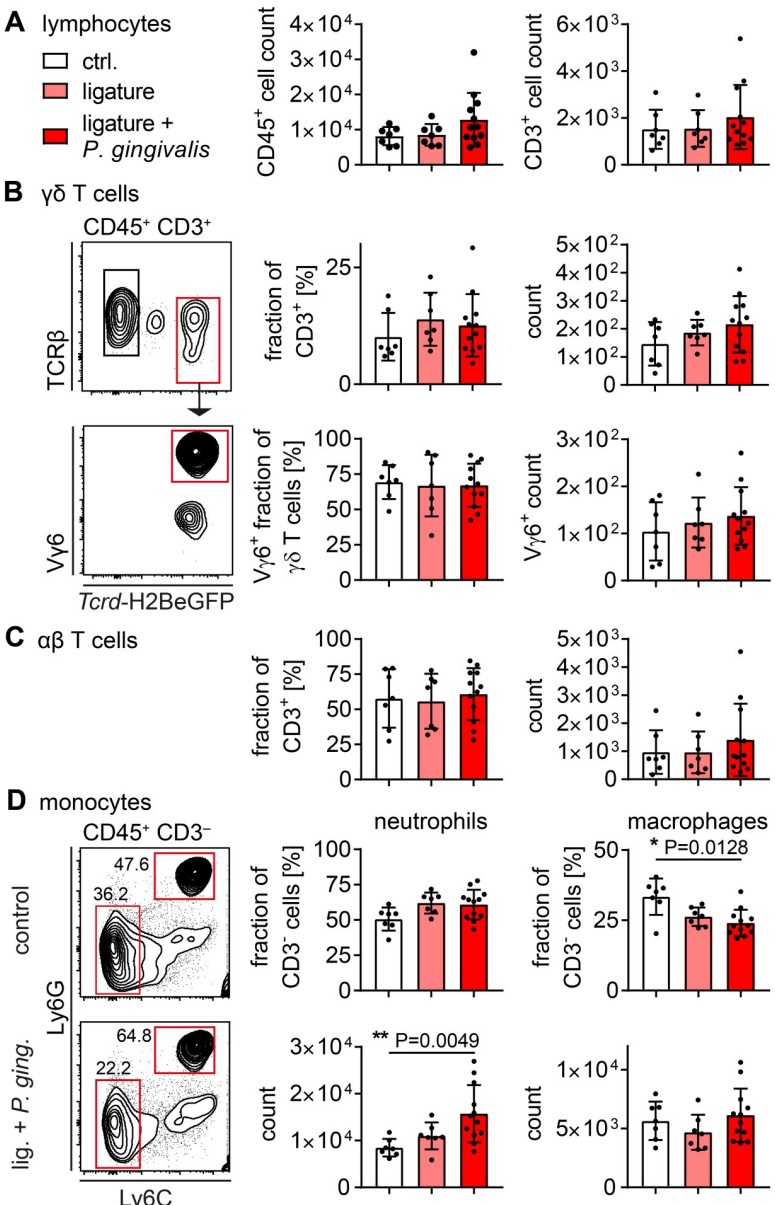

**Fig 2. Cellular immune reaction to ligature-induced periodontitis.** (A) Number of CD45[+] lymphocytes and CD3[+] T cells isolated from gingival tissue of ligature-treated and control mice. (B) Fraction and number of γδ T cells isolated from gingival tissue of ligature-treated and control mice in CD45[+] CD3[+] or CD45[+] CD3[+] TCRγδ[+] gate, respectively, and representative gating. (C) Fraction and number of αβ T cells isolated from gingival tissue of ligature-treated and control mice in CD45[+] CD3[+] gate. (D) Fraction and number of neutrophils and macrophages in CD45[+] CD3[-] gate. In all diagrams each point represents a mouse. Data were obtained from three independent experiments and statistically analyzed by Kruskal-Wallis test and Dunn´s multiple comparisons, mean and SD are shown. A p-value below 0.05 was considered as significant.

Notably, induction of IL-17 expression was only detectable in *Tcrd*-H2BeGFP and *Tcrd*[-/-] mice to a comparable level and not at all in *IL17af*[-/-] mice (Fig 3A). Apart from that, induction of IL-23 expression is diminished in *IL17af*[-/-] mice in response to the treatment (Fig 3B), which suggests in line with the missing IL-17 in these mice a weaker inflammation. Neither GAS6 nor TRAP expression was differently induced by the treatment in any group (Fig 3C and 3D), supporting our data not showing a difference in bone loss.

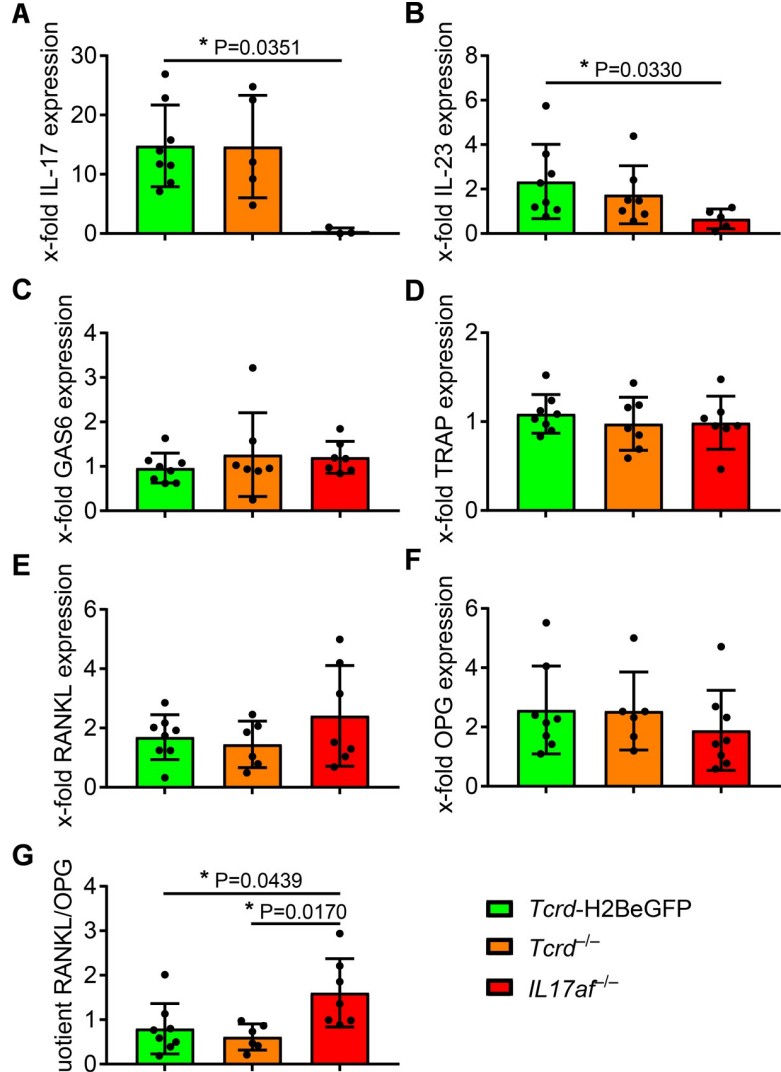

**Fig 3. qPCR shows no difference in acute osteoclast-activity between *Tcrd*-H2BeGFP, *Tcrd*⁻/⁻ and *IL17af*⁻/⁻ mice, while RANKL/OPG ratio suggests an osteoclastogenic milieu in *IL17af*⁻/⁻ mice.** The ligature-induced induction of IL-17 (A), IL-23 (B), GAS6 (C), TRAP (D), RANKL (E) and OPG (F) expression in *Tcrd*-H2BeGFP, *Tcrd*⁻/⁻ and *IL17af*⁻/⁻ mice measured by qPCR-analysis employing the $\Delta\Delta C_t$ method and RANKL/OPG ratio (G) as a measure of osteoclastogenic milieu. In all diagrams each point represents a mouse. The data are normalized to HPRT-induction by ligature-induced periodontitis. Data were obtained from two independent experiments and statistically analyzed by Kruskal-Wallis test and Dunn´s multiple comparisons, two outliers were removed by Grubb´s test, mean and SD are shown. A p-value below 0.05 was considered as significant.

Furthermore, the RT-PCR showed no difference in RANKL or OPG induction by the treatment in *Tcrd*-H2BeGFP or *Tcrd*⁻/⁻ mice (Fig 3E and 3F). In *IL17af*⁻/⁻ mice, RANKL expression seems to be slightly more induced and OPG expression slightly less induced compared to *Tcrd*-H2BeGFP and *Tcrd*⁻/⁻ mice (Fig 3E and 3F). Accordingly, a significantly increased RANKL/OPG ratio induced by the treatment occurs in *IL17af*⁻/⁻ mice (Fig 3G). Together those data suggest, there is no difference in real osteoclast activity, whereas the milieu might become in *IL17af*⁻/⁻ mice more and more osteoclastogenic, considering the elevated RANKL/OPG ratio. Apart from that, the data suggest, that the protective role of IL-17 is in the context of periodontitis largely independent of γδ T cells, as proposed before [14].

## Equal alveolar bone loss in *Tcrd*-H2BeGFP, *Tcrd*$^{-/-}$ and *IL17af*$^{-/-}$ mice after four weeks of inflammation

After no difference in bone loss in response to periodontitis in *Tcrd*-H2BeGFP, *Tcrd*$^{-/-}$ and *IL17af*$^{-/-}$ mice, there was a need to corroborate the data regarding the bone-loss-measurements, since they suggested, although statistically not significant, a tendency of diminished bone erosion not only in the *IL17af*$^{-/-}$ mice, but also in the *Tcrd*$^{-/-}$ mice. Therefore, we measured the CEJ-ABC distance on HE-stained cryo-sections. This analysis did not show any significant differences between the genotypes, whereas again the *IL17af*$^{-/-}$ mice seem to develop a less severe bone loss (Fig 4A and 4B).

Furthermore, we assessed the bone erosion in *Tcrd*-H2BeGFP, *Tcrd*$^{-/-}$ and *IL17af*$^{-/-}$ mice in response to ligature-application by μCT-imaging of the respective jaws and subsequent measurement of the CEJ-ABC distance. The 3-dimensionality of the data allowed us to assess the CEJ-ABC distance from both the palatal (Fig 4C and 4D) and the buccal side (Fig 4E and 4F) of the jaws. Applying this analysis, we could not detect any differences in bone erosion between the three genotypes (Fig 4D and 4F).

The three different imaging methods, two models and four variants of analysis we performed did not reveal any significant differences in bone loss between *Tcrd*-H2BeGFP, *Tcrd*$^{-/-}$ and *IL17af*$^{-/-}$ mice suggesting that, in line with previous reports [14], in the gingiva, the IL17-response by αβ T cells is in contrast to other epithelial sites decisive for the outcome of the inflammation. However, in contrast to earlier reports showing an amelioration of bone loss in *IL17af*$^{-/-}$ mice ten days after inflammation [14], four weeks after inflammation the alveolar bone loss of *Tcrd*-H2BeGFP and *IL17af*$^{-/-}$ mice seems to equalize.

## γδ T cells are not functionally replaced by αβ T cells in *Tcrd*$^{-/-}$ gingiva

Other experimental models suggest that γδ T cells can be functionally replaced by αβ T cells [46]. Therefore, we hypothesized that potentially Th17 αβ T cells functionally replaced the missing γδ T cells in the gingiva of the *Tcrd*$^{-/-}$ mice, hiding the expected phenotype of reduced bone loss. To test this, we applied the ligature-model to *Tcrd*-GDL mice that can be conditionally depleted of all γδ T cells by diphtheria-toxin treatment. The comparison of γδ T cell depleted and not depleted *Tcrd*-GDL mice did not reveal an amelioration of bone loss after ligature-treatment (Fig 5). This suggests that different from other sites, in periodontitis γδ T cells are not functionally replaced by αβ T cells and the equal bone-loss in *Tcrd*-H2BeGFP and *Tcrd*$^{-/-}$ mice is due to an anyway more decisive function of the αβ T cells.

## Discussion

In this study, we aimed to clarify the contradicting effects of IL-17 and its producers in the course of periodontitis. A study from Krishnan *et al.* suggested an enhanced bone-loss in *Tcrd*$^{-/-}$ mice compared to WT after ten days of inflammation, while Tsukasaki *et al.* could not see any difference between both genotypes at that time point. However, after ten days of inflammation, bone loss was reduced in *IL17af*$^{-/-}$ mice [14]. The IL-17 produced during the inflammation has been shown to be indispensable for the recruitment of neutrophils to the gingiva [27], whose efferocytosis by macrophages in turn limits the production of IL-17 [64–66]. In IL-17RA knockout mice, this recruitment is not possible and the alveolar bone loss is enhanced after six weeks of inflammation [27]. Thus, while IL-17 had a bone-damaging effect around day ten of periodontitis [14, 54], it seemed to display an overall protective role in case the inflammation persisted for a period as long as six weeks [27]. In other epithelial sites of the organism than the gingiva, IL-17 is associated with inflammation-exacerbation and in case of

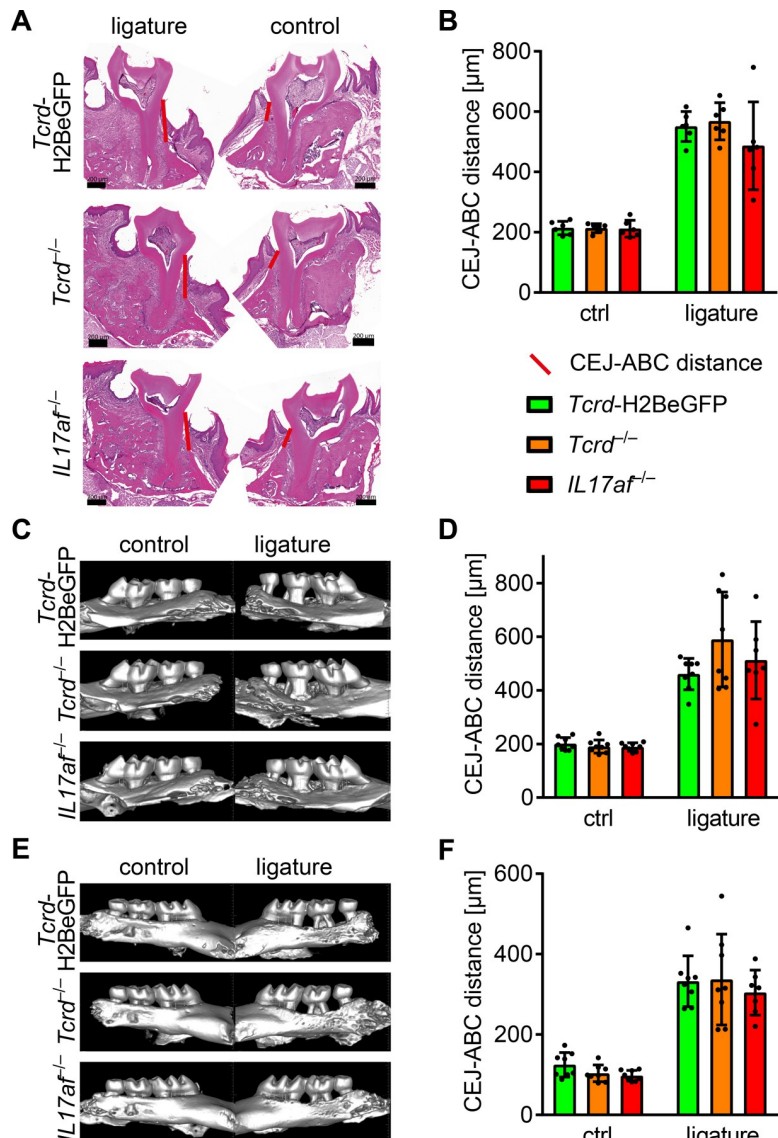

**Fig 4. No difference in ligature-induced bone loss between *Tcrd*-H2BeGFP, *Tcrd*⁻/⁻ and *IL17af*⁻/⁻ mice.**
Representative HE-stained cryo-sections (A) of jaws from ligature-treated and control *Tcrd*-H2BeGFP, *Tcrd*⁻/⁻ and *IL17af*⁻/⁻ mice and (B) measured CEJ-ABC distance. Representative μCT images of jaws from ligature-treated and control *Tcrd*-H2BeGFP, *Tcrd*⁻/⁻ and *IL17af*⁻/⁻ mice and measured CEJ-ABC distance from palatal (C, D) and buccal (E, F) side. In all diagrams each point represents a mouse. Both the HE and μCT data sets were obtained from two independent experiments and statistically analyzed by Kruskal-Wallis test and Dunn´s multiple comparisons, mean and SD are shown. A p-value below 0.05 was considered as significant. All genotypes show statistically significant bone loss compared to control, which is not depicted in the figure.

deregulation with the establishment of chronic inflammation, as for example psoriasis in the skin [7], entheseal inflammation [48] or inflammations in the lung [67]. In those sites, IL-17 production by γδ T cells plays a prominent role for induction of the inflammation and associated tissue-damaging side effects. The situation in early periodontitis reflected this effect of IL-17, although it has been shown that in the gingiva the IL-17 production by CD4⁺ αβ T cells plays an important role [68, 69] that seems to be more decisive [14] than the one by γδ T cells. Also in the human oral immune network, the gingiva was reported to accumulate immune

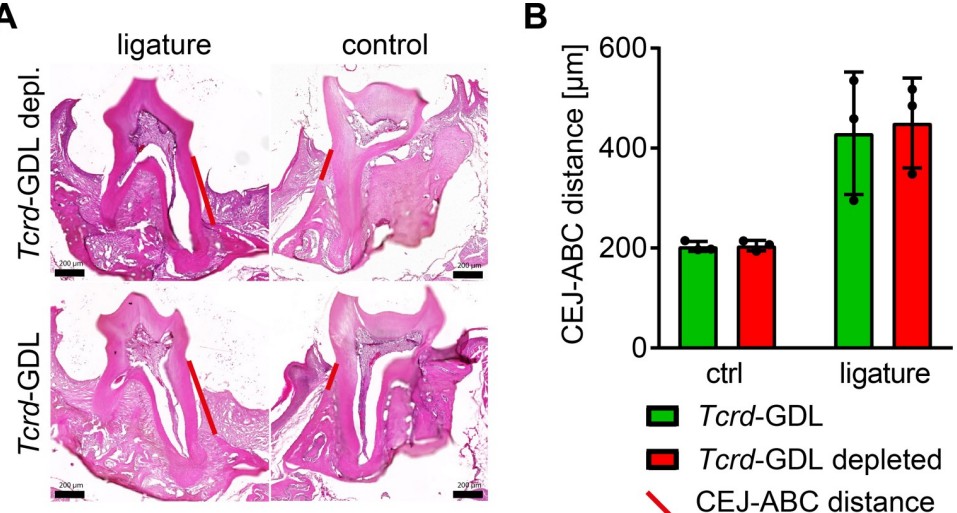

**Fig 5. Conditional depletion of γδ T cells suggests no functional replacement of missing γδ T cells by αβ T cells in the gingiva.** Representative HE-stained cryo-sections (A) of jaws from ligature-treated and control *Tcrd*-GDL and γδ T cell depleted *Tcrd*-GDL mice and (B) measured CEJ-ABC distance. Each point represents a mouse. Data were obtained from one experiment and statistically analyzed by Kruskal-Wallis test and Dunn´s multiple comparison, mean and SD are shown. A p-value below 0.05 was considered as significant. All genotypes show statistically significant bone loss compared to control, which is not depicted in the figure.

cells [70], especially neutrophils and T cells. However, while IL-17 was reported to be enhanced in human periodontitis [71–73], it was shown to be mainly produced by CD4$^+$ αβ T cells (Th17 cells) in health and periodontitis, while γδ T cells only showed a minor contribution [70]. Nevertheless, IL-17 mediates important functions for effective oral antifungal immunity also in humans [42]. Resembling this situation, our data, measured at a medium-duration of periodontitis (four weeks) showed no differences between *Tcrd*-H2BeGFP, *Tcrd*$^{-/-}$ and *IL17af*$^{-/-}$ mice. The conditional depletion of the γδ T cells, also not leading to a difference in bone loss, sustains these data. Despite that, the RANKL/OPG ratio seems to be slightly enhanced in *IL17af*$^{-/-}$ mice while osteoclast-markers also do not suggest differences in osteoclast activity.

We hypothesize that the seemingly contradictive conclusions drawn in past are caused by the different timespan between induction of periodontitis and assessment of bone-loss. From an evolutionary point of view, the teeth are very important. On the one hand you cannot eat and survive without them. On the other hand they cause a relatively fragile site in the outer body-barrier [74]. The gingiva, specifically the JE, is an extraordinary site of the body-barrier, where also the γδ T cells concentrate [8]. The JE represents a transmucosal passage in the external body-barrier connecting the bacteria inflicted oral cavity with the subjacent tissue layers including bone. Apart from that, the JE is much closer to the underlying alveolar bone than any other external body barrier epithelia. Moreover, the JE is exposed to constant mechanical forces by mastication and to the oral bacteria, which both influence the oral immune homeostasis [62, 75, 76]. While the epithelium of the oral cavity as well as other epithelial sites is protected by tight junctions, the JE is very thin and only weakly attached to the teeth via hemidesmosomes [74]. Thus, the consistency of the teeth out of mineral materials is necessary to ensure the sufficient hardness, but also makes the sealing of the connective site between tooth and gingiva difficult and weak. The reported dissemination of oral bacteria [14] to other sites of the body is reminiscent of this paradox. In other words: In case of a periodontal inflammation, which is common in humans, the organism needs to resolve the problem the fast as

possible to prohibit potential extension of the inflammation to the fragile neighboring gingiva, to minimize dissemination of bacteria in the body and to avoid tooth-loss. Our data now suggest that in absence of IL-17 after four weeks of periodontitis, the bone-damage reaches the level of *Tcrd*-H2BeGFP (i.e. wild type) mice. Therefore, we hypothesize that the function of IL-17 in early periodontitis is primarily the same as in other epithelial inflammations. It is pro-inflammatory and by this, in the unique situation in the tooth-supporting tissue, it is bone-damaging. This damage may be an evolutionary useful collateral damage, since it limits the timespan of chronic periodontitis by ultimately leading to tooth-loss and sealing of the body-barrier. Since our data suggest that in case of ongoing inflammation, bone loss in *IL17af*$^{-/-}$ becomes as intensive as in wild type mice and others showed a worsening after six weeks [27], the bone-damaging effects of IL-17 limit the timespan of bacterial dissemination to other sites, of potential spreading of the periodontitis to other tooth and potentially hampered mastication due to pain. Based on our data here, and integrating multiple prior publications using similar models, we suggest that the organism finally prefers to effectively fight the bacterial infection, at the price of tooth loss, which is the lesser evil in this context.

## Methods

### Animals

C57BL/6-Trdc$^{tm1Mal}$/J (*Tcrd*-H2BeGFP) mice [77], B6.129P2-Tcrd$^{tm1Mom}$/J (*Tcrd*$^{-/-}$) mice [78], B6.Cg-Il17a/Il17f$^{tm1.1Impr}$ (*IL17af*$^{-/-}$) mice [6] and C57BL/6-Trdc$^{tm1(eGFP-2A-hDTR-2A-CBGr99\ luciferase)Impr}$ (*Tcrd*-GDL) [46] mice were bred in the animal facility of the central animal facility at Hannover Medical School (Hannover, Germany). All animal experiments were conducted in compliance with the German animal protection law (TierSchG BGBl. I S. 1105; 25.05.1998). All animal experiments were approved by the Lower Saxony Committee on the Ethics of Animal Experiments as well as the Lower Saxony State Office of Consumer Protection and Food Safety under the permit number 33.12-42502-04-16/2167.

### *P. gingivalis* culture and preparation for infection

Culture and handling of Porphyromonas gingivalis (ATCC 33277) was performed in an anaerobic incubator under an atmosphere containing 80% nitrogen, 10% carbon dioxide and 10% hydrogen. Every other day Schaedler medium (Oxoid Limited), supplemented with vitamin K (10 μg/ml, Carl Roth GmbH & CO. KG, Karlsruhe, Germany) was inoculated with P. gingivalis to prepare liquid cultures. 6x1012 cells per animal per day were resuspended in 50 μl carboxy-methyl cellulose (CMC) 2% dissolved in PBS to prepare the inoculum for oral infection.

### Ligature application and *P. gingivalis* administration for 28 days

Prior to ligature placement all animals received a combination of ampicillin and kanamycin (2 mg each per day and animal dissolved in ultrapure water) over 5 days. The antibiotics were administered directly into the oral cavity. After two days of recovery, ligatures were applied. In brief, the mice were anesthetized (Xylacin 5 mg/kg body weight and Ketamin 100 mg/kg body weight) and a piece of suture material (PremiCron 6/0, B.Braun Surgical, S.A. Rubi. Spain) was tied around each maxillar second molar and fixed with two nods on the lingual side of the teeth. The ligature was not applied in control mice. The following 28 days, the mouth of the "ligature + *P. gingivalis*" group was flushed 5 days per week with 50 μL of bacterial inoculum, prepared as described above. The "ligature" group of mice and control mice only got CMC. At day 28, the mice were sacrificed by carbon dioxide inhalation and cervical dislocation and further processed.

## Defleshing, staining and microscopy of extracted jaws

For the quantification of alveolar bone loss around ligature treated teeth, the maxillae of the mice were partly dissected, including the complete molar area and fixed in paraformaldehyde (4%, dissolved in PBS) for 72h at 4°C. After fixation the samples were rinsed several times in PBS and stored afterwards in PBS at 4°C until further processing. To remove all the soft tissue from the jawbone and teeth, the dissected samples were incubated in 25% NaOH in a 15ml clear plastic tube (Sarstedt, Sarstedt AG & Co. KG, Nümbrecht, Germany) at room temperature and shaken vigorously several times within the tubes. When the jawbone and teeth of the maxilla were completely defleshed, they were rinsed several times in PBS, until a stable neutral pH was detectable. Afterwards they were stored in PBS until further processing. The jaws were stained in a solution of 0.5% methylene blue, dissolved in PBS, at a pH of 7.5 for 2–3 minutes. The staining facilitated the detection of the cemento-enamel junction (CEJ). Both rows of maxillary molars were photo documented from the lingual and the buccal side using a stereo-microscope (Type M3Z, Wild, Heerbrougg, Switzerland). The resulting images were analyzed using the IMS Client V16Q2 software. For the detection of alveolar bone loss the exposed area between CEJ and the alveolar bone crest was measured on the buccal and lingual side of the second molar, which had been fitted with the ligature in the experimental group.

## Histological analysis

After sacrification of the mice and removal of the jaws, the tissue was fixated over-night in 4% paraformaldehyde (PFA). The fixated jaws were then decalcified by incubating them in 0.5 M EDTA in PBS for one week, with a daily renewal of the EDTA solution. For another night the jaws were kept in 30% (mass percentage) sucrose. The fixed and decalcified tissue was embedded in Tissue Tek® O.C.T™ Compound and frozen at -20°C. The frozen blocks were cut into 10 μm sections by using a Leica cryotome CM3050, mounted on glass slides and dried for 30 min at 37°C. The sections were stained with Eosin and Hematoxylin. The imaging was performed employing a Zeiss Axio Scan.Z1 slide scanner and the resulting images were analyzed using the Zen (Blue edition) software version 3.2 from Zeiss and GraphPad Prism version 7.05.

## μCT analysis

After sacrification of the mice and removal of the maxilla, the right and left maxillary teeth-rows were excised and the gingival tissue was removed by forceps. The bone and teeth were subjected to μCT analysis, performed by the small-animal-imaging center of the central animal facility of Hannover medical school (Hannover, Germany) in a Siemens Inveon μCT (80kV, 500μA, 720 projections, binning 1; effective pixel size 8.17μm). For the analysis, the 3D-datasets were analyzed within the Inveon research workspace version 4.2 by positioning the jaws horizontally to allow a view onto the palatal or buccal side of the jaw and subsequent measurement of the CEJ-ABC distance in the software. Further data processing was performed in GraphPad Prism version 7.05.

## qPCR

After sacrification of the mice and removal of the maxilla, the right and left maxillary teeth-rows were excised and the gingival tissue was removed by forceps. The remaining bone and teeth were subjected to μCT analysis. The gingival tissue was stored in PBS and subjected to RNA-isolation by an Ultra-turrax T18 basic from IKA and Qiagen RNeasy micro-kit. 4 μg RNA were subjected to cDNA synthesis (4μg RNA in 12 μl dH2O, 1 μl oligodT (12 μM), 1 μl dNTP (2.5 mM)) and heated in a PCR-cycler for 5 min to 65°C. Immediately the mixture was

placed on ice and a PCR-mastermix (4 μl superscript III 5X FS buffer, 1 μl DTT (100 μM), 1 μl superscript III (200 U/μL), 0.1 μl RNAse out (40 U/μL)) was added. The mixture was then placed into the PCR-cycler and subjected to cDNA synthesis (60 min 50˚C followed by 15 min 70˚C). The following real-time qPCR was performed employing the SYBR® Premix Ex TaqTM II Kit from Takara Bio and HPLC-purified customized primer supplied from Sigma Aldrich (*GAS6* forward: AGGTCTGCCACAACAAACCA; *GAS6* reverse: GCGTAGTCTAATCA CGGGGG; *IL-17* forward: GCCCTCAGACTACCTCAACC; *IL-17* reverse: GTCCTAGTAGGGA GGTGTGAAGTTG; *IL-23* forward: AGCGGGACATATGAATCTACTAAGAGA; *IL-23* reverse: GTCCTAGTAGGGAGGTGTGAAGTTG; *OPG* forward: TACCTGGAGATCGAATTCTGCTT; *OPG* reverse: CCATCTGGACATTTTTTGCAAA; *RANKL* forward: TGTACTTTCGAGGGCAGATG; *RANKL* reverse: AGGCTTGTTTCATCCTCCTG; *TRAP* forward: GCTGGAAACCATGATCACCT; *TRAP* reverse: GAGTTGCCACACAGCATCAC). During the data analysis according to the $\Delta\Delta C_t$ method employing microsoft excel 2010 and GraphPad Prism version 7.05, only samples displaying $C_t$ values lower than 35 were included a priori. Data were normalized to HPRT-induction as housekeeping-gene.

## Isolation of gingival lymphocytes for flow cytometry

Isolation of gingival lymphocytes was performed as described previously [8]. In brief, after sacrification of the mice, the maxilla was carefully removed. The right and the left molar teeth-rows and the adjacent gingiva were excised with a scalpel. The tissue was subjected to a digestion by collagenase IV (2 mg/mL) and DNaseI (0.025 mg/mL) in RPMI 1640. After incubation in a shaker (1400 rpm) at 37˚C for 1h, the digestion was stopped by adding 150 μL 0.5 M EDTA and further incubation under the same conditions for 15 min. After removal of the gingival tissue from the bone and its mincing, the lymphocytes were then isolated by filtering the medium and the minced gingival tissue through a cell-strainer (100 μm), washing of the cell-strainer by 6 mL MACS buffer (3% FCS (volume percentage) and 0,8% 0.5 M EDTA (volume percentage) in PBS) and centrifugation at 1250 rpm and 4˚C for 5 min. The supernatant was removed and the cells were resuspended in 100 μL of PBS for live/dead staining (Zombie Aqua dye, BioLegend) for 5 min on ice. After washing in MACS, the cells were blocked 5 min on ice with 5% $F_c$-block (clone 2.4G2) in MACS buffer and stained with the respective antibodies (CD45.2-APC-eF780, eBioscience, clone 104; CD3e-PB, BioLegend, clone 17A2; Tcrβ-PerCP-Cy5.5, eBioscience, clone H57-597; TCRγδ, homemade, clone GL3; IgM-PE, eBioscience, clone RM-7B4; Vg5/Vg6, homemade, clone 17D1; Ly6G-Cy5, homemade, clone 1A8; Ly6C-PE-Cy7, BioLegend, clone HK1.4) diluted in MACS buffer for 30 min on ice. Flow cytometric analysis was performed employing a Becton Dickinson LSRII, FACSDiva software version 8.0.1, FlowJo version 10 and GraphPad Prism version 7.05.

## Conditional ablation of γδ T cells in vivo

The γδ T cell depletion was performed as previously described [46]. In brief, *Tcrd*-GDL mice were treated two times with 15 ng diphtheria-toxine (DTx) per gram body weight by intraperitoneal injection. The first DTx injection was performed 7 days before the ligature-application, the second one 5 days before the ligature-application. *Tcrd*-GDL mice in control groups were injected with PBS. Finally, the mice were subjected to the ligature-induced periodontitis model.

## Statistical analysis

Statistical analysis was performed in GraphPad Prism version 7.05. A p-value below 0.05 was considered as significant.

## Supporting information

**S1 Fig. Gating strategy during flow-cytometric analysis of *Tcrd*-H2BeGFP mice.** Representative gating strategy during the flow-cytometric analysis of (A) control and (B) ligature+*P. gingivalis* treated *Tcrd*-H2BeGFP mice for lymphocytes (upper panel) and neutrophils/macrophages (bottom panel).
(TIF)

**S1 Dataset.**
(XLSX)

## Acknowledgments

We thank Prof. Jörg Eberhard (Department of Prosthetic Dentistry and Biomedical Materials Science, Hannover Medical School, Hannover, Germany) for the support of the establishment of the experimental periodontitis model.

## Author Contributions

**Investigation:** Anneke Wilharm, Christoph Binz, Inga Sandrock, Francesca Rampoldi, Stefan Lienenklaus, Eva Blank, Abdi Demera.

**Project administration:** Immo Prinz.

**Supervision:** Avi-Hai Hovav, Meike Stiesch, Immo Prinz.

**Visualization:** Christoph Binz.

**Writing – original draft:** Christoph Binz.

**Writing – review & editing:** Anneke Wilharm, Christoph Binz, Stefan Lienenklaus, Eva Blank, Andreas Winkel, Meike Stiesch, Immo Prinz.

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
