## [Decision Letter · Decision Letter 0]

27 Jul 2021

PONE-D-21-18861

Interleukin-17 is disease promoting in early stages and protective in late stages of experimental periodontitis

PLOS ONE

Dear Author,

Thank you for submitting your manuscript to PLOS ONE. After careful consideration, we feel that it has merit but does not fully meet PLOS ONE’s publication criteria as it currently stands. Therefore, we invite you to submit a revised version of the manuscript that addresses the points raised during the review process.

We look forward to receiving your revised manuscript.

Kind regards,

Dr. Kunaal Dhingra, MDS, MFDS RCPS (Glasg), MFDS RCS (Eng), MDTFEd

Academic Editor

PLOS ONE

Journal Requirements:

2. Please include the method of euthanasia in the manuscript Methods.

3. As part of your revision, please complete and submit a copy of the Full ARRIVE 2.0 Guidelines checklist, a document that aims to improve experimental reporting and reproducibility of animal studies for purposes of post-publication data analysis and reproducibility: https://arriveguidelines.org/sites/arrive/files/Author%20Checklist%20-%20Full.pdf (PDF). Please include your completed checklist as a Supporting Information file. Note that if your paper is accepted for publication, this checklist will be published as part of your article.

This work was supported by the German–Israeli Foundation for Scientific Research and Development Grant 1432 (to A.-H.H. and I.P.), by a grant from the Niedersächsisch-Israelische Forschungsförderung (to A.-H.H. and I.P.), and by the DG PARO CP GABA (to A.W., I.P., and M.S.) A. Wilharm and C. Binz were supported by the Hannover Biomedical Research School.

This work was supported by the German–Israeli Foundation for Scientific Research and Development Grant 1432 (to A.-H.H. and I.P., http://www.gif.org.il/pages/default.aspx), by a grant from the Niedersächsisch-Israelische Forschungsförderung (to A.-H.H. and I.P.), and by the DG PARO CP GABA (to A.W., I.P., and M.S.) A. Wilharm and C. Binz were supported by the Hannover Biomedical Research School. The funders had no role in study design, data collection and analysis, decision to publish, or preparation of the manuscript.

5. Please note that in order to use the direct billing option the corresponding author must be affiliated with the chosen institute. Please either amend your manuscript to change the affiliation or corresponding author, or email us at plosone@plos.org with a request to remove this option.

Additional Editor Comments:

The manuscript is well written. Kindly address following concerns by reviewers:

Introduction length should be a little shorter while Discussion longer with details given in Introduction.

The last sentence of the "Discussion" element (Conclusion): "in the end IL-17 has a protective function for the organism, but a damaging for the bone" seems too general.

Reviewers' comments:

Reviewer's Responses to Questions

**Comments to the Author**

1. Is the manuscript technically sound, and do the data support the conclusions?

Reviewer #1: Yes

Reviewer #2: Yes

2. Has the statistical analysis been performed appropriately and rigorously? 

Reviewer #1: Yes

Reviewer #2: Yes

3. Have the authors made all data underlying the findings in their manuscript fully available?

Reviewer #1: Yes

Reviewer #2: Yes

4. Is the manuscript presented in an intelligible fashion and written in standard English?

Reviewer #1: Yes

Reviewer #2: Yes

5. Review Comments to the Author

Reviewer #1: As the role of IL-17 within the dental tissues is still insufficiently clarified, this study brings very interesting findings in order to provide explanation for dual effects of IL-17. This research is well designed, conducted and presented. Minor suggestion is the Introduction length (should be a little shorter), while Discussion longer with details given in Introduction.

Reviewer #2: The manuscript entitled "Interleukin-17 is disease promoting in early stages and protective in late stages of experimental periodontitis" is nicely prepared. Though, the last sentence of the "Discussion" element (Conclusion): "in the end IL-17 has a protective function for the organism, but a damaging for the bone" seems too general.

6. PLOS authors have the option to publish the peer review history of their article (what does this mean?). If published, this will include your full peer review and any attached files.

Reviewer #1: No

Reviewer #2: No

---

## [Author Response · Author response to Decision Letter 0]

26 Sep 2021

Dear Dr. Dhingra, dear reviewers,

Thank you for your careful consideration of our manuscript PONE-D-21-18861. We have the pleasure to submit a revised manuscript and to answer the questions raised by you and the reviewers.

Reply to editorial comments:

You asked to revise the manuscript regarding the PLOS one´s style requirements and file naming. We reassessed the journal-requirements and adjusted the manuscript accordingly. Since you requested to include the method of euthanasia in the methods part, we mentioned it in line 501 to 502 and as you wished, we completed the ARRIVE 2.0 Guidelines checklist. To further adhere the journal requirements, we removed the funding statement entirely from the manuscript itself. We do not want to make any changes regarding the funding statement given in the online submission form. Apart from this, we employed the Preflight Analysis and Conversion Engine (PACE) to revise the figures as you proposed.

Reply to Reviewer #1: We revised the manuscript regarding the length of the introduction and the discussion, as you suggested. We hope that the structure of the revised manuscript now meets your expectations.

Reply to Reviewer #2: We fully agree with the reviewer that this last sentence of the discussion: “in the end IL-17 has a protective function for the organism, but a damaging for the bone” is very general. According to the reviewer’s suggestion, we thus removed it from the manuscript.

Kind regards

Christoph Binz

(on behalf of all authors of PONE-D-21-18861)

---

## [Decision Letter · Decision Letter 1]

24 Jan 2022

PONE-D-21-18861R1Interleukin-17 is disease promoting in early stages and protective in late stages of experimental periodontitisPLOS ONE

Dear Dr. Binz,

Thank you for submitting your manuscript to PLOS ONE. After careful consideration, we feel that it has merit but does not fully meet PLOS ONE’s publication criteria as it currently stands. Therefore, we invite you to submit a revised version of the manuscript that addresses the points raised during the review process.

We look forward to receiving your revised manuscript.

Kind regards,

Kunaal Dhingra, MDS, MFDS RCPS (Glasg), MFDS RCS (Eng), MDTFEd

Academic Editor

PLOS ONE

Journal Requirements:

Reviewers' comments:

Reviewer #3: I found the manuscript “Interleukin-17 is disease promoting in early stages and protective in late stages of experimental periodontitis” interesting. It is well written and the methodology is rigorous.

I think that the abstract section is not quite informative. In particular lines 22-30 are a sort of “introduction section”, lines 31-33 represent “methods section”, whereas a few lines are dedicated to the results. I suggest to reduce the introductory part and pay more attention for the results obtained. Furthermore, I suggest to better specify in lines 34-36 what is intended for “bone-damaging” and “bone-protective” effects.

The study focuses on the analysis of IL-17. I believe it would be appropriate to implement the introduction and discussion sections by better investigating the role of this cytokine in the human body and in other oral disorders, especially those that are immune-mediated and / or potentially malignant.

Except for these small improvements, the topic of the study is very interesting and the effort of the authors is to be appreciated.

---

## [Author Response · Author response to Decision Letter 1]

21 Feb 2022

Dear Dr. Dhingra, dear reviewers,

Thank you for your careful consideration of our manuscript PONE-D-21-18861R1 and the offer to submit a revised manuscript and to answer the questions raised by the reviewers.

Reply to Reviewer #3: Thank you for making us aware of the mis-relation between the description of results versus other contents in the abstract. We reorganized large parts of the abstract and hope that it is now suitable for publication in PLOS ONE. Since you suggested a more intense and general description/discussion of IL-17 and its functions, we now added such sections in the introduction and discussion describing general effects of IL-17 as well as the role of IL-17 in several other diseases and models and giving a short comparison between the situation in mice and humans.

All changes in the revised manuscript are highlighted in yellow to facilitate peer review.

Kind regards

Christoph Binz

(on behalf of all authors of PONE-D-21-18861R1)

---

## [Decision Letter · Decision Letter 2]

3 Mar 2022

Interleukin-17 is disease promoting in early stages and protective in late stages of experimental periodontitis

PONE-D-21-18861R2

Dear Dr. Binz,

We’re pleased to inform you that your manuscript has been judged scientifically suitable for publication and will be formally accepted for publication once it meets all outstanding technical requirements.

Kind regards,

Kunaal Dhingra, MDS, MFDS RCPS (Glasg), MFDS RCS (Eng), MDTFEd

Academic Editor

PLOS ONE

Reviewers' comments:

Reviewer's Responses to Questions

**Comments to the Author**

I thank the authors for making the required changes. The manuscript can be accepted for publication.

---

## [Editor Report · Acceptance letter]

8 Mar 2022

PONE-D-21-18861R2 

Interleukin-17 is disease promoting in early stages and protective in late stages of experimental periodontitis 

Dear Dr. Binz:

I'm pleased to inform you that your manuscript has been deemed suitable for publication in PLOS ONE. Congratulations! Your manuscript is now with our production department. 

Kind regards, 

on behalf of

Dr. Kunaal Dhingra 

Academic Editor

PLOS ONE